# Sampling Matters in Explanations: Towards Robust Attribution Analysis with Feature Suppression

## Abstract

Pixel-wise image attribution analysis seeks to highlight a subset of the semantic features from inputs and such a subset can reflect the interactions between the features and their inferences. The gradient maps of decision risk values with respect to the inputs can highlight a fraction of the interactive features relevant to inferences. Gradient integration is a pixel-wise attribution approach by sampling multiple samples from the given inputs and then summing the derived gradient maps from the samples as explanations. Our theoretical analysis demonstrate that the alignment of the sampling distribution can delimit the upper bound of explanation certainty. Prior works leverage some normal or uniform distribution for sampling and the misalignment of their distributions can thus lead to low explanation certainty. Furthermore, their explanations can fail if models are trained with data augmentation due to the skewed distribution. We present a semi-ideal sampling approach to improve the explanation certainty by simply suppressing features. Such an approach can align with the natural image feature distribution and preserve intuition-aligned features without adding agnostic information. Further theoretical analysis from the perspective of cooperative game theory also shows that our approach is in fact equivalent to an estimation of Shapley values. The extensive quantitative evaluation on ImageNet can further affirm that our approach is able to yield more satisfactory explanations by preserving more information against state-of-the-art baselines.

## 1 Introduction

Image semantic features are the ordered pixel combinations with specific patterns agreeing with human intuitions. We use the term 'feature' as 'semantic feature' hereafter. Neural networks can summarize the features from inputs (being plural due to the emphasis on multiple piece-wise inputs) and combine the feature summaries to make decisions. Human-annotated datasets can establish the correlations between features and categories (also known as 'labels') by using human prior knowledge (Wang et al., 2018b; Vapnik, 1999). Such prior knowledge implied in training dataset is known as the prior supervision signal for guiding the learning process of neural networks (known as the 'training' process) (Mohri et al., 2018; Vapnik, 1999). The knowledge between data and categories can be statistically measured by consulting the information bottleneck principle (Tishby et al., 2000; Tishby & Zaslavsky, 2015). It is notable that the supervision signal emphasizes intuition-aligned features due to the prior knowledge comes from humans. Consequently, neural networks supervised by human-annotated datasets also underscore intuition-aligned features. This is the first intuition in this work but we will theoretically demonstrate that this statement is true in Section 3.

Pixel-wise attribution analysis – we use 'explanations' or 'pixel-wise explanations' hereafter – seeks a subset of the interactive features from inputs in which the features are relevant to decisions (See Figure 2). Highlighting the intuition-aligned features relevant to decisions is crucial for robust explanations. Yet, the non-linearity of neural networks impedes the explicit analyses regarding its complicated inference behaviours and the interactions between features and decisions. Fortunately, gradients provide an indirect device to observe the interactive features relevant to decisions. The first-order gradients of decision risk values with respect to its inputs can reflect the piece-wise importance of the inputs (Simonyan et al., 2013; Baehrens et al., 2010). High absolute gradient values indicate that the small changes at input end can lead large variations to decision scores.

In a single inference, the single derived gradient map can merely highlight a small fraction of interactive features due to the non-linearity of neural networks. This also links to the neural network saturation effect (Sundararajan et al.,

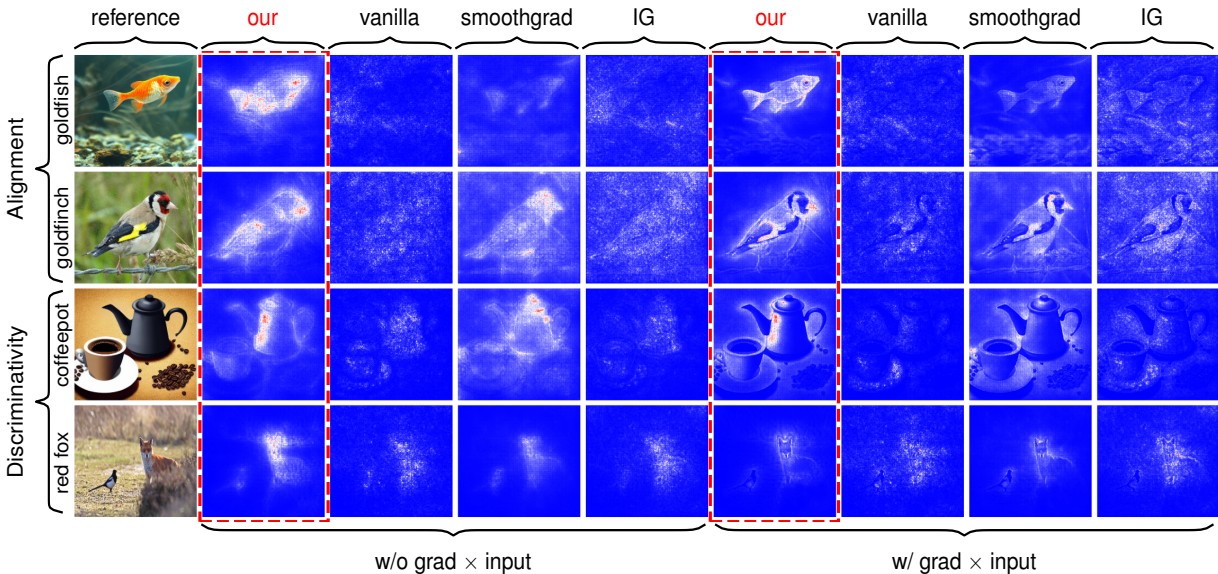

Figure 1: The figure is a performance showcase. The performances are qualitatively compared from two aspects: (1) The semantic alignment with respect to intuitions and (2) the object discriminativity with the presence of multiple objects. The evaluation model is a pre-trained ResNet50 on ImageNet. The algorithm iteration numbers are set to 50. The pixel dropping probability of our algorithm is set to 0.7. The noise level of Smoothgrad is set to 0.15 as suggested in the original paper. The explanations are normalized to $[0, 1]$ by using min-max normalization and we use the quantitative color map "bwr" ($[\text{blue} = 0, \text{white} = 0.5, \text{red} = 1]$).

2017; Krizhevsky et al., 2017; Rakitianskaia & Engelbrecht, 2015; Amit et al., 1987). Such a phenomenon does not exclude the importance of non-highlighted features – they do participate into decisions. Gradient integration is an approach to tackle this problem by integrating multiple gradient maps in which each gradient map highlights a small portion of the overall interactive features.

Prior landmark state-of-the-art works integrate multiple gradients and sample from some distributions to tackle the aforementioned feature highlighting issues. Their works usually lack further theoretical analyses. Smilkov et al. claim that adding noise can eliminate the 'noise' with agnostic origins in gradients and thus propose an approach (we cite their work as 'Smoothgrad') to create explanations by adding random noise to remove gradient noise (Smilkov et al., 2017). Nevertheless, we hold a critical view towards this approach which is not well-aligned with the feature perception mechanism of neural networks learned from supervision signals. Such a misalignment can yield implausible explanations with low certainty (See the fourth and the eighth columns in Figure 1) and lead to the sensitivity on the training perturbations (See Figure 3). Inspired by the Aumann–Shapley theory (Shapley, 1953; Roth, 1988; Aumann & Shapley, 2015), Sundararajan et al. attempt to tackle the neuron saturating problem by globally scaling inputs to create multiple samples from inputs. The derived multiple gradient maps from the samples are then integrated to create ultimate explanations (Sundararajan et al., 2017). In the remaining part of this paper, we cite their work as 'IG'. Such an approach can preserve intuition-aligned features from inputs but suffer from the lack of feature diversity. Thus, the explanations using IG remain unsatisfactory (See the fifth and the ninth columns in Figure 1).

Moreover, the aforementioned sampling approaches can also suffer from sampling saturating effect due to the lack of feature diversity through the perception lens of neural networks. Neural networks learn to extract the features relevant

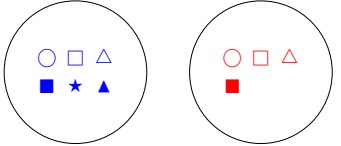

Features in inputs     Features in explanations

Figure 2: This example shows the notation of explanations. The left figure shows that the semantic features from inputs in which the features are denoted by six shapes. The right figure shows the corresponding explanation which accounts four features. The explanation is a subset of the features from inputs.

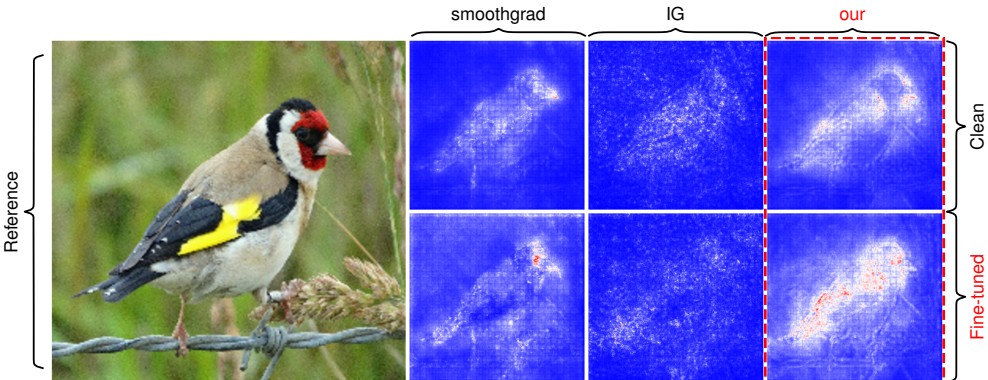

Figure 3: This experiment exhibits the failures and the sensitivity of the explanations with a fine-tuned ResNet50 on ImageNet. The model is fine-tuned with data augmentation by two transforms: (1) Randomly adding Gaussian with $\sigma$ in $[0.1, 0.3]$ and (2) randomly adjusting the luminance within $[0.1, 0.9]$. We train $5000$ batches with batch-size by $8$, learning rate $10^{-3}$ and SGD optimizer. The quality of explanations degrades due to the model learns to ignore the irrelevant perturbations from sampling.

to categories and to suppress irrelevant categories. From the perspective of information bottleneck principle (Tishby et al., 2000; Tishby & Zaslavsky, 2015), at the embedding space through the perception of neural networks, the feature diversity in embedding space accounts the amount of relevant information perceived by neural networks after filtering out irrelevant features. Experiment in Figure 5(a) shows the sample projections for various sampling approaches in which neural networks perceive them disparately. Our sampling approach aligns with IG – both are considered of well preserving intuition features. The experiment in Figure 5(b) shows the samples from a normal distribution can collapse in embedding space due to the lack of diversity through the perception lens of neural networks. The samples from normal distribution suffer either the lacking of diversity or overwhelming useful signals.

Furthermore, neural networks can learn to neglect the perturbations in the samples from the distributions not aligned with semantic feature distribution in nature images. The optimization processes drive the perception mechanism of neural networks to align with the semantic feature distribution such that the neural networks can extract the most relevant features from inputs and neglect irrelevant features. Our experiments show that fine-tuning with data augmentation can degrade and even fail explanations as models can learn to ignore the sampling perturbations while such perturbations are crucial to induce gradients. Figure 3 is a showcase of the degradation and failures of explanations with and without the presence of data augmentation (marked as "fine-tuned" versus "clean" respectively). This phenomenon corroborates the difficulty and challenges of explaining inferences. Such a phenomenon needs further holistic investigation to study the behind mechanisms. We do not unfold the discussions due to the research scope.

The above concerns in tandem with ethical concerns (See Section 8) urge us to devise an ideal and robust explanation approach with robust sampling. The content is organized in the following fashion. We first theoretically and empirically revisit the sampling problem in the context of gradient integration and derive the explanation certainty inequality from the perspectives of information theory and statistical inference. We then seeks to mitigate the sampling misalignment problem towards robust explanations by devising a semi-ideal sampling approach through simply dropping some pixels with some distribution. Our experiments suggest that our sampling approach aligns with the feature distribution in inputs. There are two theoretical paths leading to our approach – from statistical inference theory and cooperative game theory. The later theoretical analysis in Section 6 affirms that our approach is also equivalent to an estimation of Shapley value – which is the unique solution in cooperative game theory (Shapley, 1953; Roth, 1988).

## 2 Contributions

The insights in this research are not only to demonstrate an algorithm for enhancing explanation certainty but also revisit a fundamental yet crucial problem for all algorithms falling into the scope of gradient integration: Sampling matters in explanations. Our contributions are thus in three aspects: (1) The extensive revisiting of the sampling problem in the context of gradient integration with theoretical analysis and empirical study – such a delving can push the advance of the explanation research, (2) the derived quantitative approach for the assessment of explanation

certainty, and (3) the complete theoretical analysis from the perspective cooperative game theory which links our approach to an estimation of Shapley value.

## 3 Motivation

This work is largely motivated from four aspects: (1) The explanation certainty is bounded above by the alignment of sampling distribution with respect to the semantic feature distribution, (2) the form of the natural image feature distribution, (3) the low feature diversity of samples through the perception of neural networks can further delimit the explanation certainty and (4) the failures of sampling distributions with the presence of data augmentation.

### 3.1 Revisiting sampling distribution

#### 3.1.1 An upper bound of explanation certainty

Let $Pr(\boldsymbol{x}|\boldsymbol{z})$ be the probability by observing some explanations $\boldsymbol{z}$ to indirectly infer some ground truth inputs $\boldsymbol{x}$. Such a probability can reflect the explanation certainty. It is not difficult to informally show that the mutual information (Kullback, 1997; Cover, 1999) between inputs and its explanations can delimit an upper bound of $Pr(\boldsymbol{x}|\boldsymbol{z})$. Let $p(z|x)$ be the p.d.f. of some pixel $x$ in $\boldsymbol{x}$ for given some pixel $z$ in $\boldsymbol{z}$. We simplify the analysis by further assuming random variables $x$ and $z$ are i.i.d. Hence the $Pr(\boldsymbol{x}|\boldsymbol{z})$ can be approximately rewritten as:

$$Pr(\boldsymbol{x}|\boldsymbol{z}) = \prod_x \prod_z Pr(x|z) = \exp\left(\sum_x \sum_z \log Pr(x|z)\right) \approx \exp\left(\iint_{z,x} \log p(x|z) dx dz\right). \tag{1}$$

Considering:

$$\iint_{z,x} dx dz = \iint_{z,x} p(x,z) dx dz \tag{2}$$

and

$$I(\boldsymbol{x};\boldsymbol{z}) = \iint_{z,x} p(x,z) \log\left(\frac{p(x|z)}{p(x)}\right) dx dz \tag{3}$$

and

$$H(\boldsymbol{x}) = -\int_x p(x) \log p(x) dx. \tag{4}$$

Combining the above results in equations (1, 2, 3 and 4) and applying the Cauchy-Schwarz inequality:

$$\iint_{z,x} \log p(x|z) dx dz = \iint_{z,x} dx dz \cdot \iint_{z,x} \log p(x|z) dx dz = \iint_{z,x} p(x,z) dx dz \cdot \iint_{z,x} \log p(x|z) dx dz$$

$$\leqslant \iint_{z,x} p(x,z) \log\left(\frac{p(x|z)}{p(x)} p(x)\right) dx dz$$

$$= \iint_{z,x} p(x,z) \log\left(\frac{p(x|z)}{p(x)}\right) dx dz + \int_x \left(\int_z p(x,z) dz\right) \log p(x) dx$$

$$= I(\boldsymbol{x};\boldsymbol{z}) + \int_x p(x) \log p(x) dx$$

$$= I(\boldsymbol{x};\boldsymbol{z}) - H(\boldsymbol{x}) \equiv -H(\boldsymbol{x}|\boldsymbol{z}). \tag{5}$$

Hence the mutual information links to the explanation certainty by:

$$Pr(\boldsymbol{x}|\boldsymbol{z}) \leqslant \frac{\exp(I(\boldsymbol{z};\boldsymbol{x}))}{\exp(H(\boldsymbol{x}))} \equiv \exp(-H(\boldsymbol{x}|\boldsymbol{z})). \quad \square \tag{6}$$



Figure 4: This example showcase the intuition regarding semantic redundancy in nature images. Such redundancy is explained by the feature distribution in this paper. The leftmost is the reference. The four images right to the reference are with random pixel sampling ratio by 0.7, 0.6, 0.5, 0.4, 0.3 and 0.2 respectively.

This inequality associates the explanation certainty with its mutual information. In fact mutual information has more profound implications which can also be leveraged as self-supervision signals due to such a statistical connection (Hjelm et al., 2018; Kumar et al., 2019). Mutual information also sheds light on tackling the difficulty of assessing explanation quality by providing a potential faithful measure.

### 3.1.2 Gradient integration

Prior to derive the optimal sampling distribution of gradient integration. We first formulate the algorithm of gradient integration. Formally, a general gradient integration algorithm without the technique of 'inputs $\times$ gradients' (Shrikumar et al., 2017; Sundararajan et al., 2017) can be formulated as:

$$z \approx \mathop{\mathbb{E}}_{\hat{x} \sim q(x)} \left[ abs\{\nabla_{\hat{x}} f(\hat{x})\} \right] \tag{7}$$

where $x$ denotes some inputs, $z$ denotes some explanations, $\hat{x}$ denotes some sample from some sampling distribution $q(x)$, $f(\cdot)$ defines a neural network and $abs\{\cdot\}$ is a piece-wise absolute-value operator. From the aforementioned analysis in Section 3.1.1, $I(x; z)$ is determined by both the sampling distribution $q(x)$ and the property of $f(\cdot)$ – the perception lens of neural networks.

We pinpoint two factors relevant to explanation quality: (1) The alignment of the sampling distribution with respect to feature distribution, and, (2) the embedding diversity of samples from the perception mechanism of neural networks – which determines how neural networks extract features.

### 3.1.3 The optimal sampling distribution

We have shown that the mutual information determines an upper bound of the explanation certainty in equation ( 6) and also known that the general gradient integration can be formulated as equation ( 7). Further given the fact that mutual information function $I(x, z)$ is convex for given $x$, by applying Jensen's inequality:

$$\begin{aligned} I(x, z) &= I(x; \mathop{\mathbb{E}}_{\hat{x} \sim q_x} \left[ abs\{\nabla_{\hat{x}} f(\hat{x})\} \right]) \\ &\leqslant \mathop{\mathbb{E}}_{\hat{x} \sim q_x} \left[ I(x; abs\{\nabla_{\hat{x}} f(\hat{x})\}) \right] \\ &= \mathop{\mathbb{E}}_{\hat{x} \sim q_x} \left[ I(x; \hat{z}(\hat{x})) \right] \end{aligned} \tag{8}$$

where $\hat{z}(\hat{x}) := abs\{\nabla_{\hat{x}} f(\hat{x})\}$, $x$ denotes some inputs, $z$ denotes some explanation, $\hat{x}$ denotes some sample and $q_x$ denotes some sampling distribution for given some input $x$.

The ideal sampling distribution $q^*$ is an optimization problem over all inputs from some $p$ and all sampling distributions $q$:

$$q^* = \arg\max_q \mathop{\mathbb{E}}_{x \sim p_x} \left\{ \mathop{\mathbb{E}}_{\hat{x} \sim q_x} \left[ I(x; \hat{z}(\hat{x})) \right] \right\} \tag{9}$$

with the optimal solution $q^* = p$. The difficulty is that the distribution $p$ is usually unknown. We overcome this problem by using a heuristic approach to derive the $q^*$.

## 3.2 Natural image feature distribution

We simplify the feature distribution problem into a Bernoulli trial problem. Each pixel from $\boldsymbol{x}$ can present in some features with some constant probability $p$ for $\boldsymbol{x}$. In fact, a better approach is to use a neural network to predict the probabilities for images – parameterizing the sampling approach and making it learnable. But training such a neural network is difficult. We do not cover this approach in this paper. Approximately, for an image with $n$ pixels, the probability of $k$ pixels are accounted in some features – the features contain $k$ pixels – is given by a binomial distribution:

$$Pr(k, n; p) = \binom{n}{k} p^k (1-p)^{n-k}. \tag{10}$$

We leverage the approximation of this distribution to perform the sampling. This is a semi-ideal sampling approach currently since we have no knowledge to sample semantic signals directly. Figure 4 show the intuitions behind this sampling approach by dropping a portion of the pixels. Numerical experiment in Figure 6 further corroborates this distribution by measuring the gradient $L_2$ which reflects the importance sum of features from the perspective of cooperative game theory.

## 3.3 The diversity of samples

The equations (8 and 9) also imply that sampling diversity from the perception lens of neural networks can determine the explanation certainty. This is because of the learning process. The training supervision signals guides neural network to align the feature extraction with the semantic feature distribution – reinforcing the relevant features while suppressing the irrelevant features (Tishby et al., 2000; Tishby & Zaslavsky, 2015).

Consequently, one approach to evaluate the sampling quality is to project the samples from the penultimate layer of image classifiers and apply PCA projections with cosine kernel. The reason we use cosine kernel for PCA projections at the penultimate layer is because both theoretical and empirical research suggests that the Softmax layer at neural networks measures the angles rather than $L_p$ distances for inferences.

It is not difficult to show that the statement is true. Let $\boldsymbol{z}$ be the inputs at Softmax layer – also be the outputs from the penultimate layer, $\boldsymbol{w}_i$ be the weights with respect to the i-th class, $s_i$ be the i-th class score and assume vectors are normalized with $L_2$. The score for the i-th class before the exponential operation is given by $s_i = \boldsymbol{z}^T \boldsymbol{w}_i = \cos\theta$ where $\theta$ is the intersection angle between $\boldsymbol{z}$ and $\boldsymbol{w}_i$. This suggests image classifiers with Softmax layer learns to align the embedding vectors with respect to class weight vectors. This fact also motivates state-of-the-art works in facial recognition models based on metric learning (Kulis et al., 2013) such as CosFace (Wang et al., 2018a) and ArcFace (Deng et al., 2019).

In the experiments of Figure 5(a), we measure feature diversities by computing cosine similarities at the penultimate layer and visualize the projections using PCA with cosine kernel. The results show that our samples align with the samples from IG and disalign with samples from noise subspace. Samples from IG can be used as reference to measure the deviation of samples from intuitions. The results imply that the sampling distribution by Smoothgrad does not align well with the feature distribution in semantic feature signals. Neural networks can fail to extract useful features from samples if samples come from a narrow state space or do not align with the semantic feature distribution. The experiments in Figure 5(b) show such a feature collapse phenomenon.

## 3.4 The failures of explanations

Neural networks learn to align its perception mechanism with the semantic feature distribution due the training supervision signals are from humans. The training and fine-tuning processes can learn to account the perturbations from non-aligned distributions – e.g. normal distribution – as irrelevant noise and ignore them. But it is notable that neural networks can not learn to ignore the samples from the distributions aligned with semantic feature distributions since the signals are relevant to labels. Figure 3 show the example that the explanations can degrade and fail with a fine-tuned pre-trained ResNet50 on ImageNet. This phenomenon urges us to seek a robust sampling approach which can align with the semantic feature distribution and the perception mechanism of neural networks.

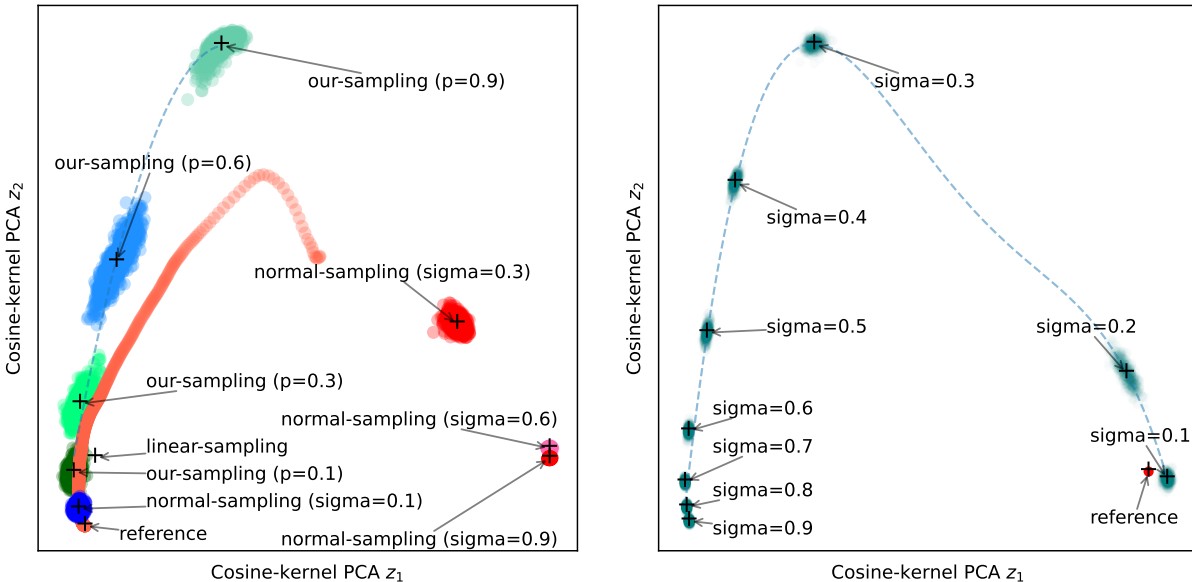

(a) The 2D feature projections at the penultimate layer of a pre-trained ResNet50.

(b) The feature perception collapse for normal-sampling at the penultimate layer of a pre-trained ResNet50.

Figure 5: In the left figure, we project the feature vectors from the penultimate layer of a pre-trained ResNet50 on ImageNet using PCA with cosine kernel. We collect 1000 samples for each sampling approach. We vary the noise level of normal-sampling (used in Smoothgrad) from $0.1$ to $0.9$. We vary the pixel dropping probability of our-sampling approach from $0.1$ to $0.9$ as well. We use "+" to indicate the projection centers. In the right figure, we project the feature vectors for the samples from normal-sampling with 1000 samples. We vary the noise level from $0.1$ to $0.9$. The trajectory shows the feature space collapse effect that noises ultimately overwhelm signals and neural networks fail to extract patterns as the noise levels increase.

## 4 Sampling with feature suppression

Natural images contain considerable redundant information. The semantic signals in images are not susceptible to the absence of some pixels. Figure 4 shows that when a majority of the pixels are absent the features in the samples remain discernible to humans for making faithful inferences by using the remaining signals. Our theoretical analysis in Section 3.2 regarding the feature distribution can further confirm this fact. In fact, the later empirical study (See Figure 6) further confirms our sampling approach aligns with semantic feature distribution by measuring the $L_2$ norm with respect to the ratio of the absence of pixels.

Feature suppression refers to randomly replace some pixels with a constant $C$ ($C \approx 0$) at a sampling process. The $C$ plays the role to suppress some piece-wise inputs. The amount of the presence of pixels follows some binomial distribution in equation (10). The sampling can thus be broken down into two phases: (1) Sampling the amount of pixels $k$ from some binomial distribution and (2) randomly choosing $k$ pixels from images as a sample, and, the absent pixels will be replaced with the constant $C$. For the sake of numerical stability in some cases, the constant $C$ is set to a small value ($C = 10^{-8}$) instead of zero. Otherwise, the explanations may have checkerboard patterns for some models.

### 4.1 The optimal sampling probability $p$

Yet, there is a final question to answer: The optimal sampling $p$ in the devised distribution. We conduct empirical experiments to measure the coalition (a set of pixels) contribution distribution for answering this question. The $L_2$ norm of gradients of samples can indirectly reflect the importance of the coalition contributions on some decision risk function if we vary the coalition sizes. Let $f(\boldsymbol{x})$ be the decision risk function and $\boldsymbol{x}$ be inputs. The decision risk

perturbations due to the variations at inputs are approximated by using Taylor expansion for $f(\boldsymbol{x} + \Delta\boldsymbol{x})$ at $\boldsymbol{x}$:

$$||f(\boldsymbol{x} + \Delta\boldsymbol{x}) - f(\boldsymbol{x})||_2 = ||\Delta\boldsymbol{x}^T\nabla_{\boldsymbol{x}}f(\boldsymbol{x}) + \mathcal{O}(\Delta\boldsymbol{x})||_2 \approx ||\nabla_{\boldsymbol{x}}f(\boldsymbol{x})\Delta\boldsymbol{x}||_2 \leqslant ||\nabla_{\boldsymbol{x}}f(\boldsymbol{x})||_2 \cdot ||\Delta\boldsymbol{x}||_2. \tag{11}$$

where the $L_2$ norm of gradients can reflect the decision risk variations with respect to inputs.

In the experiment (as shown in Figure 6), we measure the $L_2$ norms of the gradients from multiple images by varying the coalition sizes at a pre-trained ResNet50. Each light pink line is corresponding to an image and the blue line is the averaged $L_2$ norm. The result shows that the feature contributions from the perception lens of neural networks reach to large values when the pixel dropping probability varies from $0.5$ to $0.8$. In our experiments, the explanation quality achieves best performance when this probability is between $0.5$ and $0.8$.

## 5    Related works

We have revisited the gradient integration based approaches (Smoothgrad and IG) from the perspective of sampling aspect in Section 1 and Section 3. We conduct a brief literature review herein to further introduce indirectly relevant works from the following two categories.

**Local approximation based approach**.   LIME (Local Interpretable Model-Agnostic Explanations) is a model-agnostic but local explanation approach which assumes a linear model is more understandable and uses a linear model to approximate the study model for understanding the decision behaviours (Mishra et al., 2017). SHAP (SHapley Additive exPlanations) is also a model-agnostic explanation approach and decomposes predictions on the basis of the inner product of Shapley value vector and coalition vector (Lundberg & Lee, 2017). The Shapley values can thus be approximately estimated by learning. Their approach is fundamentally different from our approach in that our work approximately derives Shapley values by using gradients as a proxy from study model directly and integrating the collected gradients as explanations.

**Activation decomposition based approach**. DeepLIFT (Deep Learning Important FeaTures) compares the activation of each neurons to their references and assign values accordingly for deriving explanations (Shrikumar et al., 2017). LRP (Layer-wise Relevance Propagation) decomposes outputs backwards according to their weights to input end as explanations (Montavon et al., 2019). CAM (Class Activation Map) leverages the spatial correlation between inputs and activation maps in CNN networks, sums the activation maps from chosen layers – e.g., the ultimate layer before classification network – with the weights with respect to class scores as explanations. (Li et al., 2018). Grad-CAM incorporates activation map approach with gradient information to further improve explanation quality as opposed to CAM (Selvaraju et al., 2017).

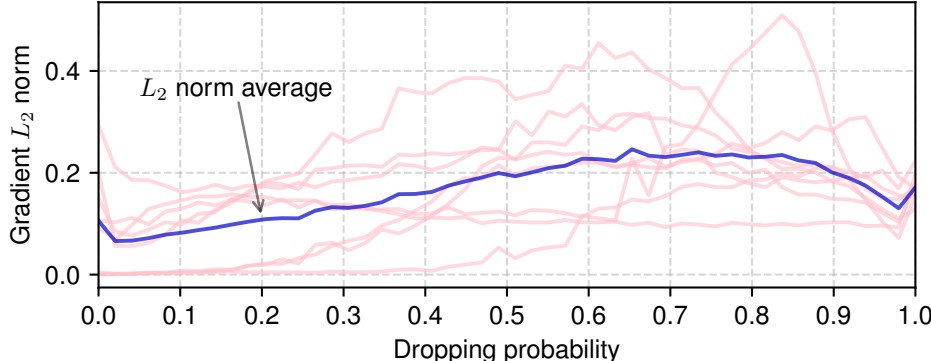

Figure 6: The experiment shows the coalition (feature) contributions on the decisions with respect to coalition sizes (controlled by the pixel dropping probability). We measure the gradient $L_2$ for multiple images by varying the coalition sizes at a pre-trained ResNet50. Each light pink line is corresponding to an image. The blue line is the average over all images. The result agrees with our theoretical results in both statistical analysis and cooperative game theory.

# 6 Theoretical analysis

In this section, we conduct a holistic theoretical analysis from the perspective of cooperative game theory (Winter, 2002; Shapley, 1953; Roth, 1988) to justify our algorithm: The gradient integration with feature suppression. We show that such an algorithm is equivalent to an estimation of Shapley values. In an image classifier, each image pixel can be viewed as a pixel player in a cooperative game which is set by such an image classifier. All pixel players play the cooperative game and their cooperations lead to decision scores (known as payoffs). Such a setting naturally falls into the scope of cooperative game theory.

We revisit the discrete version of Shapley value theory and link the theory to our approach for laying a solid theoretical guarantee to the fairness of our algorithm. The fairness is crucial for coherent explanations. Shapley value theory aims to address the fair payoff distribution problem in cooperative games (Branzei et al., 2008; Davis & Maschler, 1965; Driessen, 2013). It is notable that Shapley value is the unique solution in cooperative game theory.

## 6.1 Fair distribution in Shapley value theory

Suppose a cooperative game with some player set $N$ and some characterization function $v$. The characterization function $v$ maps a group of players (known as coalitions) from the power set $\mathbb{P}(N)$ of $N$ to a real number (known as payoff). A Shapley value function $\phi_j(v)$ gives the fair distribution – which satisfies with a set of desirable axioms – from some payoff to some player $j$ by summing the marginal contributions over all coalitions from the power set $\mathbb{P}(N)$:

$$\phi_j(v) := \sum_{S \subset \mathbb{P}(N) \backslash \{j\}} \frac{1}{|N| \cdot \binom{|N|-1}{|S|}} \Big[ v(S \cup \{j\}) - v(S) \Big] \tag{12}$$

where $S$ is a subset of the power set $\mathbb{P}(N)$ without the presence of some player $j$ and the $|\cdot|$ operator gives set cardinality.

## 6.2 Coalition distribution

In fact, a coalition can be viewed as a semantic feature. The coalition distribution agrees with the analysis of feature distribution of natural images in Section 3.2.

For some coalition $S$ from $\mathbb{P}(N) \backslash \{j\}$ without the presence of some player $j$, the number of the coalitions with the size $|S|$ is thus given by $\binom{|N|-1}{|S|}$. The number of all possible coalition sizes is $|N|$. The probability of some coalition $S$ is given by:

$$Pr\{S\} := \frac{1}{|N| \cdot \binom{|N|-1}{|S|}}. \tag{13}$$

The Shapley value equation (12) can be rewritten into the form of statistical expectation by using the coalition distribution in equation (13):

$$\phi_j(v) = \sum_{S \subset \mathbb{P}(N) \backslash \{j\}} Pr\{S\} \Big[ v(S \cup \{j\}) - v(S) \Big]$$
$$= \underset{Pr\{S\}}{\mathbb{E}} \Big[ v(S \cup \{j\}) - v(S) \Big]. \tag{14}$$

## 6.3 Linking Shapley value to gradients

The loss values in an image classifier can be accounted as the payoffs of the cooperative game set by the image classifier. The loss function is a candidate of the characterization function of such cooperative game. However, the loss function does not satisfy with the zero-payoff principle for empty coalition such that the empty coalition set $\varnothing$ shall give value $0$.

Instead, we construct a characterization function $f(\cdot)$ from the loss function $\mathbb{L}(\cdot, \cdot)$ by:

$$f(\boldsymbol{x}) := \mathbb{L}(\boldsymbol{x}, y) - \mathbb{L}(\varnothing, y) \tag{15}$$

where $\boldsymbol{x}$ denotes some pixel coalition, $y$ denotes some ground truth label and $\varnothing$ denotes zero vector. In practice, zero denotes the absence of pixel players and non-zero indicate the presence of pixel players.

Set $v := f$ and plug the equation (15) into the equation (14), the Shapley value function $\phi_j(f)$ for the j-th player with some characterization function $v := f$ can be approximated by applying Taylor expansion:

$$
\begin{aligned}
\phi_j(f) &= \underset{\hat{\boldsymbol{x}} \sim Pr\{\boldsymbol{x}\}}{\mathbb{E}} \Big[ f(\hat{\boldsymbol{x}}_{+j}) - f(\hat{\boldsymbol{x}}_{-j}) \Big] \\
&= \underset{\hat{\boldsymbol{x}} \sim Pr\{\boldsymbol{x}\}}{\mathbb{E}} \left[ \frac{\partial f}{\partial \hat{x}_j} \Delta \hat{x}_j + \frac{1}{2} \frac{\partial f^2}{\partial^2 \hat{x}_j} (\Delta \hat{x}_j)^2 + \mathcal{O}(\Delta \hat{x}_j) \right] \\
&\approx x_j \underset{\hat{\boldsymbol{x}} \sim Pr\{\boldsymbol{x}\}}{\mathbb{E}} \left[ \frac{\partial f}{\partial \hat{x}_j} \right]
\end{aligned}
\tag{16}
$$

where $\hat{\boldsymbol{x}}$ denotes some sample from $Pr\{\boldsymbol{x}\}$ for some $\boldsymbol{x}$, $\hat{\boldsymbol{x}}_{+j}$ denotes the sample with the presence of the j-th pixel in $\hat{\boldsymbol{x}}$ (set $\boldsymbol{x} := \hat{\boldsymbol{x}}_{+j}$), $\hat{\boldsymbol{x}}_{-j}$ denotes the sample with the absence of the j-th pixel in $\hat{\boldsymbol{x}}$, and $x_j$ denotes the j-th pixel value in sample $\hat{\boldsymbol{x}}$. Let $\Delta \hat{x}_j$ be the delta of the j-th pixel value in $\hat{\boldsymbol{x}}$ and know $\Delta \hat{x}_j = x_j$. We collect all piece-wise Shapley values and write the equation (16) into the form of matrix:

$$
\phi_{\boldsymbol{x}}(f) \approx \boldsymbol{x} \odot \underset{\hat{\boldsymbol{x}} \sim Pr\{\boldsymbol{x}\}}{\mathbb{E}} [\nabla_{\hat{\boldsymbol{x}}} f] = \boldsymbol{x} \odot \underset{\hat{\boldsymbol{x}} \sim Pr\{\boldsymbol{x}\}}{\mathbb{E}} [\nabla_{\hat{\boldsymbol{x}}} \mathbb{L}]
\tag{17}
$$

where the operator $\odot$ denotes Hadamard product (Horn, 1990).

This approximation links Shapley value theory with gradient integration. Shapley value adheres to a set of desirable properties: 'Efficiency', 'Symmetry', 'Linearity' and 'Null player' (Shapley, 1953; Roth, 1988) and guarantees the fair distribution in cooperative game theoretically.

The equation (17) underscores the prerequisite for robust explanations from the perspective of cooperative game: **If and only if (iff) the samples are sampled from the feature distribution (coalition distribution)**. This result echos our simplified analysis in Section 3.2.

### 6.4 Estimating Shapley values

Using Monte Carlo method (James, 1980; Metropolis & Ulam, 1949; Hammersley, 2013), the Shapley value function $\phi_{\boldsymbol{x}}(v)$ can be estimated by:

$$
\phi_{\boldsymbol{x}}(f) \approx \frac{\boldsymbol{x}}{K} \odot \sum_{k=1}^{K} \nabla_{\hat{\boldsymbol{x}}_k} \mathbb{L}.
\tag{18}
$$

The assumption for such numerical estimation is that sampling distribution aligns with the coalition distribution. In practice, we estimate an upper boundary of the $\phi_{\boldsymbol{x}}(f)$ instead. Since $abs\{\cdot\}$ is convex and use Jensen's inequality piece-wisely:

$$
\phi_{\boldsymbol{x}}(f) \leqslant abs\{\phi_{\boldsymbol{x}}(f)\} \leqslant \frac{\boldsymbol{x}}{K} \odot \sum_{k=1}^{K} abs\{\nabla_{\hat{\boldsymbol{x}}_k} \mathbb{L}\}.
\tag{19}
$$

## 7 Information leakage in batch inference

From the perspective of cooperative game, the inferences with batch sizes larger than one can have unintended feature interactions from other images in the batches. This phenomenon is known as the information leakage problem in batch inferences and relevant algorithms such as batch-ed normalization (Wu & Johnson, 2021; Ioffe & Szegedy, 2015). This problem affects explanation qualities due to the information leakage. We advise small batch sizes or single instance in inferences for deriving high quality explanations.

We give the error bound of the explanation certainty with batch inferences. Let $\boldsymbol{z}_i$ be some explanation for some inputs $\boldsymbol{x}_i$, $\boldsymbol{x}_j$ be other inputs in the same batch where $j \neq i$ and $K$ be the batch size. The explanation certainty is bounded

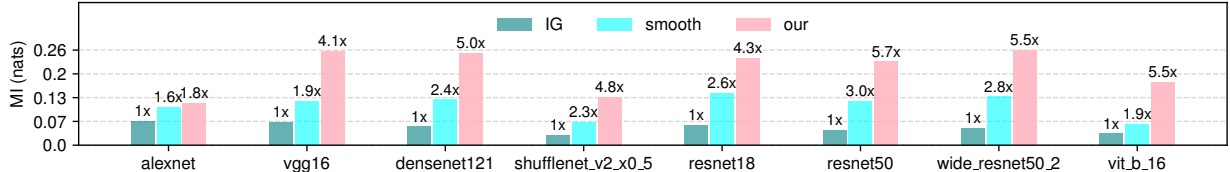

Figure 7: We extensively evaluate our algorithm against the gradient integration baselines over ImageNet dataset. Due to computation resource, we only choose landmark works as baselines – Smoothgrad and IG. All random seeds are set to $1860867$ to guarantee the reproducibility. We choose the prevailing models which are pre-trained on ImageNet. We assess the semantic alignment between explanations and semantic annotations by measuring the mutual information with bins set to $20$. For each run, we use $100$ epochs and $100$ images. The results can show two insights: (1) Our algorithm can outperform state-of-the-art landmarks and (2) the results can also indicate the robustness of models.

above by:

$$Pr(\boldsymbol{x}_i|\boldsymbol{z}_i, \boldsymbol{x}_1, \boldsymbol{x}_2, \cdots, \boldsymbol{x}_{i-1}, \boldsymbol{x}_{i+1}, \cdots, \boldsymbol{x}_K) \leqslant \exp\left(I(\boldsymbol{x}_i; \boldsymbol{z}_i) - H(\boldsymbol{x}_i) - \sum_{j \neq i} I(\boldsymbol{x}_j; \boldsymbol{z}_i)\right) \tag{20}$$

$$\leqslant \exp(I(\boldsymbol{x}_i; \boldsymbol{z}_i) - H(\boldsymbol{x}_i)). \tag{21}$$

This inequality shows that batch inference can also degrade explanation certainty. It is not difficult to show that the relative explanation certainty error is bounded above by:

$$\epsilon_i \leqslant \frac{|Pr(\boldsymbol{x}_i|\boldsymbol{z}_i) - Pr(\boldsymbol{x}_i|\boldsymbol{z}_i, \boldsymbol{x}_1, \boldsymbol{x}_2, \cdots, \boldsymbol{x}_{i-1}, \boldsymbol{x}_{i+1}, \cdots, \boldsymbol{x}_K)|}{Pr(\boldsymbol{x}_i|\boldsymbol{z}_i)}$$

$$= 1 - \exp\left\{\sum_{j \neq i} I(\boldsymbol{x}_j; \boldsymbol{z}_i)\right\}. \tag{22}$$

Ideally, a single instance inference will have higher explanation certainty.

## 8 Ethical Impact

The great success of the implementations of neural networks from a variety of applications has been beheld and heralds a new era. However, the ethical issues caused by the failures of sampling misalignment in explanations are neglected. Such a sampling misalignment in the explanations of inferences can lead ethical issues if explanations are no longer faithful. We conduct this research to tentatively understand such an impact and tackle this problem. We hope our research can urge the concerns and shed light on the ethical issues.

## 9 Evaluation

We use both qualitative evaluation approach to compare the performance against baselines. We choose three baselines: Smoothgrad, IG and vanilla due the research scope and computation resource. We only choose the landmark works in gradient integration. We do not choose the variants of gradient integration such as Grad-CAM for this sake. The implementation is available at github.

The performances are evaluated from two aspects: (1) Semantic alignment with single object and (2) object discriminativity amid the presence of two objects. The semantic alignment evaluation aims to verify if algorithms can yield explanations agreeing with intuitions. The object discriminativity aims to examine if algorithms can highlight correct objects agreeing with ground truth when multiple objects present.

### 9.1 Qualitative showcase

In Figure 1, we showcase the results with and without the 'Gradient × Input' technique. We choose ResNet50 (pre-trained on ImageNet) as the showcase evaluation model. The showcase shows that the results from our algorithm can

align with intuitions for all reference images. However, the baselines fail to highlight objects agreeing with intuitions for some cases. We do not show large-scale qualitative results as its subjectivity and difficulty.

## 9.2 Quantitative evaluation

We also an extensive quantitative evaluation regarding semantic alignment by measuring the mutual information as we derive from theoretical analysis. The mutual information can be used to compute explanation confidences theoretically by using the equation (6). We choose landmark models from AlexNet, VGG, ResNet to Visual Transformer (ViT) to perform an extensive evaluation. The mutual information unit is in *nats* and the results are also normalized with respect to the IG on the model basis. The images are randomly chosen by fixing random seed for reproducibility on ImageNet.

The evaluation results show that our algorithm can achieve higher confidences against baselines. There are two insights from the results: (1) Sampling matters in explanations and (2) some state-of-the-art models such as ResNet and ViT are more robust compared with prior models.

## 10 Conclusions and further work

This work theoretically and empirically revisits the sampling problem in the context of gradient integration and links our approach to cooperative game theory. The analyses and experiments justify and echo our claims: (1) Sampling matters in explanations and (2) robust explanations need robust sampling. Yet, further questions remain open to answer: (1) Seeking the ideal sampling, (2) the failures of explanations, and (3) the impact on explanations from information leakage in inferences. We hope to conduct further research to answer the relevant intriguing open questions.

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

## A   Appendix

## B   Algorithm

The devised algorithm using information suppression sampling approach is shown as in Algorithm 1. We use a truncated Poisson distribution as the approximation of the feature binomial distribution.

---

**Algorithm 1:** The proposed algorithm using information suppression sampling approach.

---

**Input**   : *model*, *input*, *label*, *criterion*, *prob*=0.6, and *K*=200
**Output:** $explanation$

1  $explanation \leftarrow 0$
2  $N \leftarrow$ the number of pixels in $input$
3  $C \leftarrow 10^{-8}$
4  **for** $k = 1$ **to** $K$ **do**
       /* Sampling a coalition size                                                    */
5      $\lambda = N \cdot prob$
6      $num = truncated\_poisson\_sampling(\lambda, N)$
       /* Randomly setting pixels to constant C                                  */
7      $\hat{x} = random\_set\_pixels(input, num, C)$
8      $pred = model(\hat{x})$
9      $loss = criterion(pred, label)$
10     $loss.backward()$
11     $explanation+ = loss.grad.abs()$
12 **end**
13 $explanation = (\frac{x}{N}) \cdot explanation$
14 $explanation = min\_max\_norm(explanation)$

---

