# OpenReview forum: "Sampling Matters in Explanations: Towards Robust Attribution Analysis with Feature Suppression"
_TMLR — Rejected by TMLR_

### Review · Reviewer_PjEh · 2022-12-15

**Summary Of Contributions:**

The paper explores the sampling method used for gradient-based explanation techniques.
In particular, the paper proposes to sample features to suppress rather than adding Gaussian noise.
The paper explores different pixel dropping probabilities and makes some loose connections to Shapley-based explanations.
The paper proposes an explanation metric called explanation certainty and tries to optimize this measure by the choice of the smapling distribution.
The paper gives qualitative results comparing to vanilla gradient, Integrated Gradients, and SmoothGrad.


**Audience:**

Yes

**Broader Impact Concerns:**

The paper includes a short "Ethical Impact" section but does not have much content or depth. This section should be updated with possible concerns if your method is not faithful, e.g., making operators overly confident in their model because the explanations look better. This could have adverse affects. This is where


**Claims And Evidence:**

No

**Requested Changes:**

See weaknesses above.


**Strengths And Weaknesses:**

*Strengths*

- The paper investigates problems with prior gradient-based explanation techniques.

- The paper proposes a feature dropping method based on some theoretical bounds.

*Weaknesses*

- (Critical weakness) The derivation in Sec 3.1.1 seems wrong. The cauchy-Schwarz inequality is for L2 norms but it looks like you are trying to apply it to L1 norms. (Perhaps you are looking for the Holder inequality?)
    - Also, Equation 2 doesn't make sense. The double integral needs to be over some function so the left-hand side doesn't make sense. I think you are just saying that it is 1.

- (Critical weakness) Why does "confidence"/"uncertainty" of the explanation matter, i.e., why does the mutual information perspective promote better explanations? Why not directly optimize the explanation for this metric?  It seems that it would just try to reconstruct the image. A simple algorithm is just to set z=x and the mutual information will be maximized. Thus, this metric seems very problematic
  - The distribution of $Pr(x|z)$ is also not well-defined. Is $x \in \{0,1,2,\cdots,255\}$, i.e., a pixel value? Or is it avalue between 0 and 1?  This is confusing because each pixel is actually a vector of RGB values.
  - The paper just says "Such a probability can reflect the explanation certainty." This is far from a sufficient explanation.
  - Overall, this is a fundamental problem with the paper and calls into question the usefulness of all the results.
  - Other metrics in the field, e.g., [Yeh et al. 2019] and followup works, should be used to evaluate the proposed method.

Yeh, C. K., Hsieh, C. Y., Suggala, A., Inouye, D. I., & Ravikumar, P. K. (2019). On the (in) fidelity and sensitivity of explanations. Advances in Neural Information Processing Systems, 32.

- (Major weakness) Terminology is not well-defined including "semantic features", "interactive features", "explanation (un)certainty", "natural image feature distribution", and "intuition-aligned features". More generally, much of the writing lacks clarity.
  - Most of these need to be defined explicitly using formal notation. For example, what is the "natural image feature distribution"? What are "semantic" features? How would you know you had a "semantic" feature versus a "non-semantic" feature?  What would be a non-interactive feature?  How do you quantify "intuition-aligned features"? It seems that many of these would require human experiments.

- "Further theoretical analysis from the perspective of cooperative game theory also shows that our approach is in fact equivalent to an estimation of Shapley values." - The Shapley value approach only works if you use the coalition distribution. However, your approach does not seem to enforce this coalition distribution or perhaps it upper bounds these feature values. In fact if you use a fixed $p$ for the Binomial distribution, then it will not be the coalitional distribution. Thus, it does not have the same properties. Even if the Shapley values are bounded, this will no

- Evaluation seemed insufficient to support claims. The evaluation section was only 1 page with a single qualitative figure and one quantitative measure, which was introduced in the paper.
   - "The evaluation results show that our algorithm can achieve higher confidences against baselines. There are two insights
from the results: (1) Sampling matters in explanations and (2) some state-of-the-art models such as ResNet and ViT
are more robust compared with prior models." - Both of these results
   - See above about the evaluation metric used. Adding other standard evaluation metrics would improve the paper.

- The batch inference leakage part seems out-of-place and should likely be removed or moved to the appendix. It seems that the BatchNorm layers should just be turned off during explanation (e.g., `model.eval()` in PyTorch).

---

> ### Author Response · Authors · 2022-12-23
> **Regarding: The derivation in Sec 3.1.1 seems wrong**
>
> Thanks for the very careful, useful and constructive comments. Much appreciated.
>
> Problem: (Critical weakness) The derivation in Sec 3.1.1 seems wrong. The cauchy-Schwarz inequality is for L2 norms but it looks like you are trying to apply it to L1 norms. (Perhaps you are looking for the Holder inequality?)
>
> Response: I can confirm the problem you pointed out is true. But the result can still be deducted by using other techniques.
>
> Problem: Also, Equation 2 doesn't make sense. The double integral needs to be over some function so the left-hand side doesn't make sense. I think you are just saying that it is 1.
> Response: I can confirm it is true. The LHS should be 1.
>
> I will put the revisions (more rigorous and formal) into next revision. I'm also examine other deductions in this paper. Thanks again. Very useful comments.

---

> ### Author Response · Authors · 2022-12-23
> **Regarding: Why does "confidence"/"uncertainty" ...**
>
> This part confused several reviewers, I think in the revision I need to depict the settings to clarify and define this problem formally.
>
> Intuitively, explanation certainly is a problem like this: If we give machine some x, machine outputs some prediction and also outputs an explanation z tells us this is what the machine thinks what are important in x. For humans, if we present z, what is the possibility humans think z actually links to x.
>
> Before discussing the Pr(x) or Pr(z) We need to define what is the probability here.
>
> Simply put, if x = (x1, x2, ... xn) is an image (rearranging pixels from 2D or 3D to 1D), and assume pixels are discrete (0-255 for instance) -- in fact, continuous values lead to same conclusions.
>
> P(x_i) defines the pixel probability of the i-th pixel has the value x_i over some set X -- for example the entire natural images or a subset of natural images.
>
> P(x_i) comes from some distribution P_X. We further assume pixels are i.i.d.
>
> For some image x, the image probability is a joint probability of all pixels P(x).
>
> Now, we can define what is explanation probability for z.
>
> We hope models can line with human intuitions in terms of inferences -- perceiving similar robust features to make the similar inferences. If this statement is true, we hope the outputted explanation z also follow a similar iid distribution as P(x).
>
> If some machine outputs an explanation z for x, we can consider this problem: For fixing that z, what is the probability of x given z P(x|z) -- for example, if some machine outputs an explanation z, and presents to a human and asks, how similar the z to x?
>
> Intuitively, good explanation should give higher P(x|z). If the machine outputs some z, but the P(x|z) is low, we can conclude the inference done by machine has low-quality.
>
> This is the setting.

---

> ### Author Response · Authors · 2022-12-23
> **Regarding the terminology definitions.**
>
> (Major weakness) Terminology is not well-defined including "semantic features", "interactive features", "explanation (un)certainty", "natural image feature distribution", and "intuition-aligned features". More generally, much of the writing lacks clarity.
> Most of these need to be defined explicitly using formal notation. For example, what is the "natural image feature distribution"? What are "semantic" features? How would you know you had a "semantic" feature versus a "non-semantic" feature? What would be a non-interactive feature? How do you quantify "intuition-aligned features"? It seems that many of these would require human experiments.
>
> We also consulted other papers in XAI community and model robustness research community, including theoretical papers. Most papers simply take these notions informally without formal definitions. In our paper, we use the terms in this way:
>
> * Semantic features:  Features are just patterns from inputs. In terms of the term "features", papers in adversarial vulnerability community perhaps heavily use this term and well-defined. For example, in the theoretical works [1] and [2].
>
> * Interactive features: There are so many features, but only a portion of them meaningful to neural networks. From the learning dynamics (behaviours), neural networks learn some way to extract/use features -- more specifically if annotations are provided by humans, neural networks will align with the feature uses of humans (this can be empirically verified and we did this in another paper talking about "feature perception" evolution in training).
>
> For example, if we input noise into neural networks, the prediction probabilities (if use classifier as an example) don't have obvious fluctuations, because NN doesn't "think" the features in "noise" are "meaningful". Thus, the patterns are not expressed here. But if we input a flower image to NN, the NN will give some peak in the prediction vector.
>
> * "Intuition-aligned": If give a human a cat image to ask what is it, human use intuitions to give the answer by observing the semantic-features: the head, tail, legs, etc.. If give same image to NN, but if NN use the caption under the cat image to make the inference, the feature perceptions are not aligned.
>
> [1] Andrew Ilyas, Shibani Santurkar, Dimitris Tsipras, Logan Engstrom, Brandon Tran, and Aleksander Madry. Adversar- ial examples are not bugs, they are features.
> [2] Dimitris Tsipras, Shibani Santurkar, Logan Engstrom, Alexander Turner, and Aleksander Madry. Robustness may be at odds with accuracy.

---

> ### Author Response · Authors · 2022-12-23
> **Regarding the Shapley value theory.**
>
> We revisit the discrete atomic version of Cooperative game. Shapley value is a solution concept to this game.
>
> We do use the coalitions to evaluate the marginalised gains. The "coalitions" are naturally formed by dropping some pixels.
>
> It is not possible to evaluate all coalitions, this is why sampling is very important too.
>
> In fact, IG and Smooth and many gradient-based approaches can be derived from the Shapley value solution directly. The major difference is the sampling approach in the equations are different.

---

> ### Author Response · Authors · 2022-12-23
> **Why sampling important to explanations and fixing p.**
>
> Assumptions:
> P1:  NN uses features to make decisions --- this is "common sense" but VERY IMPORTANT for statement P2.
> P2: If important are absent int some sample x' for original x by sampling. The predictions should have large variations indicting NN "thinks" it's important.
>
> The equation derived from Shapley value/cooperative game theory needs the samples over all possibilities regarding "coalition". But it's technically difficult to do so. Because we don't have the knowledge regarding all "coalitions". Instead, we can use a heuristic way by looking into the sizes of coalitions instead of coalitions. This may lead to "importance sampling" problem. We ignore this by only estimating an upper bound as explanations -- since we changed the sampling distribution which needed by Shapley value theory.
>
> But sampling by fixing also makes sense. If the samples have features absent and these features are important to the decisions, the neural networks should give some large variations in loss values. The variations also reflect on gradients. By measuring the variations with L_2 we can see NNs "use" the features carried in around 30% of pixels. This is why we use 0.7 as the parameter.

---

### Review · Reviewer_afjL · 2022-12-19

**Summary Of Contributions:**

This paper studies the issues with existing saliency map methods due to problems with sampling, and proposes a new method with relationship to cooperative game theory that is shown to perform better on a mutual information metric.

**Audience:**

Yes

**Broader Impact Concerns:**

Current discussion is adequate.

**Claims And Evidence:**

No

**Requested Changes:**

See weaknesses.

**Strengths And Weaknesses:**

Strengths:

1. This paper covers an important problem: generating good explanations for black-box predictions.

2. This paper considers a wide range of gradient-based explanation methods.

Weaknesses:

1. Writing. This paper reads rather weird as an ML paper, and in many places hard to understand. For example, I couldn't fully understand the the first sentence in Introduction, and the second sentence seems obvious to an audience with ML background. I would suggest thorough revision to the writing, with the help of readers working on ML research. I would also suggest the authors to refer to current ML papers (e.g., those referenced in the paper and the reviews) to get a feeling of the general writing style and detailedness.

2. What are the "without grad x input" and "with grad x input" variants of IG? IG implements a path integral, and I don't think has two modes of computation. If by "without grad x input" the authors meant Eq. 7, then it is simply wrong. This is not how IG works.

3.  The evaluation is rather weak:

    a. The qualitative examples of Fig. 1 don't convincing show the proposed method as a better than the rest.

    b. Only a mutual information result is given as quantitative evidence. The authors should implement many additional evaluations, such as sanity check [1], proxy metrics [2] and ground truth calibration [3].

    c. There are also non-gradient-based methods like LIME, SHAP, meaningful perturbation, RISE. These methods are also strong baselines that should be compared against.

4. I don't understand the information leakage claim in batch norm in Sec. 7. At inference time, the batch norm should be switched to the evaluation mode (e.g., ```.eval()``` in pytorch), and the batch size should not matter at all. If it matters, then that's a bug in the code.

[1] https://arxiv.org/abs/1810.03292

[2] https://arxiv.org/abs/1509.06321

[3] https://arxiv.org/abs/2104.14403

---

> ### Author Response · Authors · 2022-12-23
> **Regarding the writing.**
>
> Thanks for commenting our work, much appreciated.
>
> Regarding: "Image semantic features are the ordered pixel combinations with specific patterns agreeing with human intuitions."
>
> We refer the notion "semantic features" in this way: Features are the patterns in inputs, patterns refer the combinations of pixels with some order.
>
> Regarding: "Neural networks can summarize the features from inputs (being plural due to the emphasis on multiple piece-wise inputs) and combine the feature summaries to make decisions."
>
> Thanks for commenting this. It's true. This is a "common sense". But from the learning dynamics, it has very profoud implications which lead to our motivation. We assume the feature perceptions of neural networks learn to align with humans if annotations are provided by humans. For example, human use the information "eyes, tail, legs" to recognize the image is a cat, neural networks should behave in the same way. In my another paper, I did empirical experiments and demonstrated this statement is perhaps true.
>
> Regarding: I would suggest thorough revision to the writing, with the help of readers working on ML research. I would also suggest the authors to refer to current ML papers (e.g., those referenced in the paper and the reviews) to get a feeling of the general writing style and detailedness.
>
> Thanks for the suggestions, improving the writing quality is important.

---

> ### Author Response · Authors · 2022-12-23
> **Regarding: What are the "without grad x input" and "with grad x input" variants of IG? IG implements a path integral, and I don't think has two modes of computation. If by "without grad x input" the authors meant Eq. 7, then it is simply wrong. This is not how IG works.**
>
> Thanks for commenting.
>
> In the visualisation of the explanations regarding IG or Smooth, one technique is usually used is called "grads \times inputs".
> Earlier researchers found that simply multiplying the explanations (summed gradients for both IG and smoothgrad) with input images can improve the visualisation quality.  Please refer the depiction "Some techniques create a final sensitivity map by multiply- ing gradient-based values and actual pixel values (Shriku- mar et al., 2017; Sundararajan et al., 2017). " from [1].
>
>
> More comments regarding "grads x inputs"
> This technique actually has very profound reasons from our theoretical analyses:
> 1. Using cooperative game theory and Shapley value theory, the analytical solution for the explanations is in the form "grads x inputs" -- see the theoretical analysis.
> 2. This result can also be deducted using numerical approximation technique (Taylor expansion for example). Because the first-order term of the Taylor expansion in the form of "first-order gradients \times inputs"..
>
>
>
> [1] Smoothgrad: https://arxiv.org/pdf/1706.03825.pdf

---

> ### Author Response · Authors · 2022-12-23
> **Regarding 3: The evaluation problem (a), (b) and (c)**
>
> The evaluation is rather weak:
> a. The qualitative examples of Fig. 1 don't convincing show the proposed method as a better than the rest.
> b. Only a mutual information result is given as quantitative evidence. The authors should implement many additional evaluations, such as sanity check [1], proxy metrics [2] and ground truth calibration [3].
> c. There are also non-gradient-based methods like LIME, SHAP, meaningful perturbation, RISE. These methods are also strong baselines that should be compared against.
>
>
> Regarding (a):
> Gradient-based explanation approach itself is not novel and indeed has a long history since saliency map. This paper actually does not aim to seek/design a SOTA approach outperforming previous SOTAs -- IG and Smoothgrad and other variants for example. To be honest, the works in this area in terms of the first-order gradients has been almost done. As the paper title suggested: Sampling matters, we want to revisit the explanation quality problem in gradient-based approaches.
>
> Our approach is better for some cases, but worse for some cases. In terms of the mutual information, our approach is slightly better than Smoothgrad.
>
>
> The following comments from the response for Reviewer rkBK can partially address the concern (a):
>
> In fact, we aim to theoretically analyse the notion 'explanation quality' or 'explanation certainty' problem -- which is not done yet by other literatures -- only for gradient-based approach. But the insights can be used to evaluate other explanation works and can be impactful to XAI community. There are some metrics regarding XAI explanations: For example, human judges, PG (point game), perturbation based, etc. But they have some limits.
>
> We formulate the explanation quality into a probability problem. Intuitively, the explanation quality is a problem from the human aspect: ** If we input a x into a machine, the machine gives prediction and also gives a so-called explanation z. ** When humans examine z, humans should predict some connection between z and x. This is a conditional probability problem P(x|z).
>
> We found that the 'explanation certainty' links to mutual information (in the form of an upper bound). Moreover, the distributions of samples affect the mutual information. The logic is:
>
> ** Formulating explanation quality/certainty --> Mutual information is an upper bound in the exponential form --> Sampling distribution affects the mutual information
>
> At the end of this paper, we also suggest a simple way to improve the distribution alignment to improve mutual information.

---

> ### Author Response · Authors · 2022-12-23
> **Regarding the information leakage.**
>
> Thanks for pointing it out by setting .eval().
>
> If setting .eval(), there is nothing to worry about.
>
> For algorithms having certain inter-instance interactions, for instance, inputing 4 samples A,B,C,D as a batch [A, B, C, D].
> If network use them to compute some parameters for instance, let assume t = T(A, B, C, D), and network later uses t in inferences.
> The problem is called information leakage problem, because the explanation A' of A has some information from other instances. I formulated this problem and gave an upper bound.  Batch normalisation is only ONE aspect on information leakage. Any process relating the inter-instance interactions can have information leakage problem. BTW: batch normalisation can be turned by .eval() or set batch-size to one. I think it's also an important aspect regarding "explanation quality" problem.

---

### Review · Reviewer_rkBK · 2022-12-22

**Summary Of Contributions:**

This paper revisits the research problems of sampling algorithms in designing the model explainer based on feature-importance. Authors have provided theoretical analysis on their proposed method and some empirical studies for evaluation. The results show that the proposal could work as a replacement of Shapley values.

**Audience:**

Yes

**Claims And Evidence:**

Yes

**Requested Changes:**

Please address my comments in the weakness section.

**Strengths And Weaknesses:**

Strength:
* This paper theoretically and experimentally revisits the sampling issue in designing the feature-importance algorithms.
* The direction is novel and interesting.
* Some theoretical analysis shows that the proposed method is equivalent to Shapley values

Weakness:
* I don't really see the sampling problem of smoothgrad and ig. Maybe authors need to emphesize the issue with more intuitive figure and introduction.
* Section 3&4 are not very easy to follow. Leading to the result that the theory is not very persuadable. I don't understand the first sentence in Section 3.1. Pr(x|z) means that given the explanation, we can infer the original inputs? It seems strange. Could authors make clear about this part? Assupmtions that x and z are iid are also very strong. Neighbor pixels are surely strongly dependent of each other. Authors should give more detailed discussions about this.
* The Evaluation part seems not finished? 9.2 Quantitative evaluation does not show any results. The experiment setups and evaluation metrics are not presented. The Qualitative results are not very persuadable, in my opinion.

Minors:
* Regarding Figure 5, "we measure feature diversities by computing cosine similarities at the penultimate layer and visualize the projections using PCA with cosine kernel". Do authors compute the cosine similarities among 1000 samples to obtain a 1000x1000 matrix, and then use PCA to project the 1000x1000 matrix to 1000x2? Why not use the features 1000x4096 and project to 1000x2 directly?
* What are landmarks? How these are evaluated?

---

> ### Author Response · Authors · 2022-12-23
> **1. Regarding the two comments: "I don't really see the sampling problem of smoothgrad and ig. Maybe authors need to emphesize the issue with more intuitive figure and introduction."**
>
> Much appreciated for the very careful and constructive comments.
> Huge thanks.
>
> Response 1:
> Gradient-based explanation approach itself is not novel and indeed has a long history since saliency map. This paper actually does not aim to seek/design a SOTA approach outperforming previous SOTAs -- IG and Smoothgrad and other variants for example. To be honest, the works in this area in terms of the first-order gradients has been almost done.
>  * The first-order gradients can only reflect a part of the first-order interactions in features (regarding the term "feature", we prefer the definition "derived from patterns in data distributions" [1]).
>
> In fact, we aim to theoretically analyse the notion 'explanation quality' or 'explanation certainty' problem -- which is not done yet by other literatures -- only for gradient-based approach. But the insights can be used to evaluate other explanation works and can be impactful to XAI community. There are some metrics regarding XAI explanations: For example, human judges, PG (point game), perturbation based, etc. But they have some limits.
>
> We formulate the explanation quality into a probability problem. Intuitively, the explanation quality is a problem from the human aspect:
> ** If we input a x into a machine, the machine gives prediction and also gives a so-called explanation z.
> ** When humans examine z, humans should predict some connection between z and x. This is a conditional probability problem P(x|z).
>
> We found that the 'explanation certainty' links to mutual information (in the form of an upper bound). Moreover, the distributions of samples affect the mutual information. The logic is:
>
> ** Formulating explanation quality/certainty --> Mutual information is an upper bound in the exponential form --> Sampling distribution affects the mutual information
>
> At the end of this paper, we also suggest a simple way to improve the distribution alignment to improve mutual information.
>
> [1] Andrew Ilyas, Shibani Santurkar, Dimitris Tsipras, Logan Engstrom, Brandon Tran, and Aleksander Madry. Adversar- ial examples are not bugs, they are features.

---

> ### Author Response · Authors · 2022-12-23
> **Regarding the probability definition etc.**
>
> 2. Regarding the following comments:
> Section 3&4 are not very easy to follow. Leading to the result that the theory is not very persuadable. I don't understand the first sentence in Section 3.1. Pr(x|z) means that given the explanation, we can infer the original inputs? It seems strange. Could authors make clear about this part? Assupmtions that x and z are iid are also very strong. Neighbor pixels are surely strongly dependent of each other. Authors should give more detailed discussions about this.
>
> Response:
> Thanks for pointing this out.
> We agree with you, thanks again.
>
> The writing at this section is not clear and confusing.
>
> Before conducting the analyses, it's better to describe the settings.
> In the revision, we hope to fix this problem and deliver the content in a clearer way.
>
> Simply put, if x = (x1, x2, ... xn) is an image (rearranging pixels from 2D or 3D to 1D), and assume pixels are discrete (0-255 for instance) -- in fact, continuous values lead to same conclusions.
>
> P(x_i) defines the pixel probability of the i-th pixel has the value x_i over some set X -- for example the entire natural images or a subset of natural images.
>
> P(x_i) comes from some distribution P_X. We further assume pixels are i.i.d.
>
> For some image x, the image probability is a joint probability of all pixels P(x).
>
> Now, we can define what is explanation probability for z.
>
> We hope models can line with human intuitions in terms of inferences -- perceiving similar robust features to make the similar inferences. If this statement is true, we hope the outputted explanation z also follow a similar iid distribution as P(x).
>
> If some machine outputs an explanation z for x, we can consider this problem: For fixing that z, what is the probability of x given z P(x|z) -- for example, if some machine outputs an explanation z, and presents to a human and asks, how similar the z to x?
>
> Intuitively, good explanation should give higher P(x|z). If the machine outputs some z, but the P(x|z) is low, we can conclude the inference done by machine has low-quality.
>
> This is the setting.
>
> We hope to add more details into the revision.

---

> ### Author Response · Authors · 2022-12-23
> **Regarding the comment "The Evaluation part seems not finished? ..."**
>
> Thanks for pointing out.
>
> We put the qualitative showcase to intuitively showcase how different the explanations from vanilla, Smoothgrad, IG and our approach can be. Large scale qualitative evaluations are difficult because it needs human judges involved.

---

> ### Author Response · Authors · 2022-12-23
> **Regarding the comment about the "feature diversities" ...**
>
> Comment:
> Thanks for commenting this part.
>
> In terms of projections, we use sk-learn's PCA with kernel to do so. Let say for some embeddings with dimension 512, if we collect 1000 samples, we have a matrix 1000 x 512. This matrix is fed into PCA with cosine kernel to get the projections 1000x2.
>
> We also did 3D projections. The results are more interesting. The 3D projections show how the "geometric structures" of the features derived from the samples -- from our approach, Smmothgrad and IG -- differ from each other. The results show that the "geometric structures" of our samples are aside IG samples -- more human-intuition aligned. Perhaps in next revision, we replace the 2D with 3D projections.
>
> We're interested in the "geometric/topological" structures of "feature representations". We hope samples are representative: Their diversities should be high enough.
>
> Intuitively, from the perspective of learning dynamics, if models learn the "knowledge" implied in the "data-target" relevance (providing supervision signals) very well. The models will:
>
> Learn to align with the feature uses implied in supervision signals. For examples, if the humans who annotated datasets use intuitions to annotate, instead of noticing the captions of images. Neural networks will learn this intuitions.
> In fact, in an experiment (from my paper, not published yet) regarding the learning dynamics, the learning patterns correlate with the annotation origins and are disparate. I hope I can put the experimental results here soon after published.
> The features should be discriminative enough to make inferences. If the "features" are out-of-distribution of the supposed feature structures, the measured similarities should be very close, because models tend to ignore them without "activations".
> If we sample samples from some x, and the samples have the absence of some features, the deviations from each other should be high enough. This means the "sampling" approach can "hit" on "important aspect" regarding features. Sampling is only a "device/proxy/way" to verify what features are important to inferences, what features are not, in explanations.
>
> For such a sake, sample diversities (not the diversities among data, the diversities refer to the embeddings) from the perspective of neural networks are important. This aspect doesn't appear in other papers.
>
> Another intuitive understanding regarding sampling from for instance Gaussian noise is, in high-dimensional space (e.g. >100), for any given x' from N(u, d), an counter-intuitive property is x' is almost orthogonal to x. By adding the x' into x, perhaps, can hardly "push" or "reshape" the "geometric structure" of features in x. This problem actually also exists in other explanation works such as LIME.
>
> What are landmarks? How these are evaluated?
> We refer "landmark(s)" as the most important works in SOTAs, for instance, IG and Smoothgrad. There are so many variants from them, but they are not counted as "landmark" works. It is not possible to evaluate them all. We only focus on the discussion "Sampling matters" by evaluating the IG (uniformly sampling) and Smoothgrad (Gaussian).

---

### Author Response · Authors · 2022-12-23
**Further considerations.**

Thanks for the constructive friendly comments which are so helpful for us to improve the research quality and make it more sound.

We'll now work in the revisions according to the comments to make our manuscript more sound.

The author(s).

---

### Decision · Action_Editors · 2023-02-02

**Recommendation:** Reject

**Comment:**

Despite the interest in the topic addressed, the reviewers have pointed multiple major weaknesses in this work:
- issues with the mathematical formulations
- restricted evaluations
- lack of motivation for the problem of uncertainty

The authors were not able to provide a satisfactory response to these points.

**Audience:**

At this stage, this work does not carry sufficient evidence to be of interest to TMLR readership.

**Claims And Evidence:**

This work addresses uncertainty in model explanations. However, they do not propose a convincing argument or motivation as to why this aspect is important for robust explanations. Correct mathematical derivations and thorough evaluations are lacking.